# Molecular Pathways Potentially Involved in Hallucinatory Experiences During Sleep Paralysis: The Emerging Role of β-Arrestin-2

**DOI:** 10.3390/ijms26157233

**Published:** 2025-07-26

**Authors:** Lena M. Rudy, Michał M. Godlewski

**Affiliations:** 1Scientific Society of Veterinary Medicine Students, Warsaw University of Life Sciences—SGGW, Nowoursynowska 159, 02-776 Warsaw, Poland; s222313@sggw.edu.pl; 2Department of Physiological Sciences, Warsaw University of Life Sciences—SGGW, Nowoursynowska 159, 02-776 Warsaw, Poland

**Keywords:** β-arrestin 2, sleep paralysis, hallucinations, 5-HT2A receptor, E/I balance, oscillatory stability, serotonin

## Abstract

Sleep paralysis (SP), an REM parasomnia, can be characterized as one of the symptoms of narcolepsy. The SP phenomenon involves regaining meta-consciousness by the dreamer during REM, when the physiological atonia of skeletal muscles is accompanied by visual and auditory hallucinations that are perceived as vivid and distressing nightmares. Sensory impressions include personification of an unknown presence, strong chest pressure sensation, and intense fear resulting from subjective interaction with the unfolding nightmare. While the mechanism underlying skeletal muscle atonia is known, the physiology of hallucinations remains unclear. Their complex etiology involves interactions among various membrane receptor systems and neurotransmitters, which leads to altered neuronal functionality and disruptions in sensory perception. According to current knowledge, serotonergic activation of 5-hydroxytryptamine-receptor-2A (5-HT2A)-associated pathways plays a critical role in promoting hallucinogenesis during SP. Furthermore, they share similarities with psychedelic-substance-induced ones (i.e., LSD, psilocybin, and 2,5-dimethoxy-4-iodoamphetamine). These compounds also target the 5-HT2A receptor; however, their molecular mechanism varies from serotonin-induced ones. The current review discusses the intracellular signaling pathways responsible for promoting hallucinations in SP, highlighting the critical role of β-arrestin-2. We propose that the β-arrestin-2 signaling pathway does not directly induce hallucinations but creates a state of network susceptibility that facilitates their abrupt emergence in sensory areas. Understanding the molecular basis of serotonergic hallucinations and gaining better insight into 5-HT2A-receptor-dependent pathways may prove crucial in the treatment of multifactorial neuropsychiatric disorders associated with the dysfunctional activity of serotonin receptors.

## 1. Introduction

The mechanism underlying the atonia of striated skeletal muscles that occurs during SP is well characterized. It involves the hyperpolarization of motor neurons through the activation of both metabotropic GABA_B receptors and ionotropic GABA_A/glycine receptors [1,2,3]. In contrast, the signaling pathways promoting hallucinogenesis are not yet fully characterized and remain the focus of ongoing research.

Serotonergic hallucinations (visual and auditory) emerging during episodes of sleep paralysis often include mystical experiences (such as the perception of higher beings or out-of-body experiences), while meta-consciousness (self-reflection, detachment from the experience, attentional control) is preserved [4,5,6]. These differ from dopaminergic hallucinations—so-called “lifelike” or realistic hallucinations—typically observed during psychotic episodes in schizophrenia [7]. Genetic and environmental factors, dysregulation of acetylcholine signaling pathways, and dysfunction of mirror neurons contribute to the complex etiology of hallucinations. However, a critical condition for hallucinogenesis in sleep paralysis appears to be atypical alterations in the activation of 5-HT2A receptors, which subsequently lead to disturbances in the functioning primarily of pyramidal neurons as they exhibit the highest expression of 5-HT2A receptors in the neocortex [8]. The consequence of these disruptions is the altered integration of information from various brain regions, ultimately affecting both sensory perception and consciousness. Another key factor necessary for the manifestation of hallucinations may be atypical levels of 5-HT, the endogenous ligand of 5-HT2A receptors [9].

To better understand the signaling pathways dependent on 5-HT2A receptor activation and their role in the etiology of hallucinations during sleep paralysis, this article examines possible functional analogies in receptor behavior by comparing different ligands interacting with the receptor. This approach is significant as numerous molecular, pharmacological, and neuroimaging studies have indicated the essential role of the 5-HT2A receptors in the induction of mystical states and/or out-of-body experiences (OBEs) observed after the use of psychedelic substances [10,11,12,13,14]. On this basis, a relationship has been established between 5-HT2A receptor activation and the perceptual changes and temporary distortions of consciousness induced by psychedelics. Examples of such substances include naturally occurring compounds like psilocybin (4-PO-DMT) and mescaline (3,4,5-trimethoxyphenethylamine), as well as synthetic compounds such as lysergic acid diethylamide (LSD) and 2,5-dimethoxy-4-iodoamphetamine (DOI). Despite these ligands’ ability to bind to the 5-HT2A receptor—similarly to 5-HT—their differing chemical structures determine the activation of distinct combinations of intracellular signaling pathways [9,15]. The ability of specific ligands to induce particular intracellular signal transduction profiles appears to be a key factor in eliciting hallucinogenic effects. The possible intracellular signaling routes and their analogies to 5-HT action, as well as potential causes of 5-HT-dependent hallucinogenic effects in sleep paralysis, will be discussed in detail in subsequent sections.

## 2. Review

### 2.1. Characteristics of Serotonergic Hallucinations

The phenomenology of hallucinatory experiences during SP may vary depending on context. While the occurrence, organization, and selection of these experiences are grounded in neurophysiological mechanisms, cultural factors embedded in consciousness also appear to play a significant role. These cultural underpinnings can emotionally influence the experience, thereby shaping its content and direction. Serotonergic hallucinations—triggered by serotonin or pseudo-triggered by psychedelic substances—have been grouped together due to their mystical character (clearly distinguishing them from dopaminergic “lifelike” hallucinations) and because of their shared molecular target: the 5-HT2A serotonin receptor. Although the qualitative descriptions of experiences reported by individuals undergoing sleep paralysis are not identical to those after ingesting psychedelic substances, some important similarities do exist, such as the presence of complex visual imagery, a distorted sense of body ownership or boundaries, intense emotional salience, and the perception of otherworldly entities or presences. From a molecular physiology standpoint, this raises the question of how strongly the serotonergic system is involved in promoting such specific experiences. Secondly, what are the implications of serotonin-dependent transmission at the level of global neuronal network functionality? And, thirdly, how does this impact the atypical processing of sensory information in brain regions responsible for integrating both internal and external inputs? The aforementioned atypical processing may manifest as the hallucinations observed during SP. To characterize serotonergic hallucinations, this section focuses on hallucinatory states in SP and those observed following ingestion of LSD (as a model psychedelic compound).

REM sleep accompanying SP encompass a wide range of altered consciousness phenomena [3,16,17,18]. A key factor in promoting hallucinatory experiences seems to be the nature of REM sleep itself, during which neural activity approaches or even surpasses that of wakefulness [1]. In specific brain regions—including the pons, occipital cortex, and lateral geniculate nucleus (LGN)—neural activity has been shown to intensify markedly, in some cases surpassing levels observed during full conscious wakefulness [1] (see Figure 1). The intensified neuronal dynamics during this phase are closely associated with the occurrence of the most vivid, emotionally charged, and narratively complex dreams. Qualitative analyses of these experiences reveal that, regardless of cultural background (often rooted in early folkloric belief systems) or geographic origin, respondents who have experienced SP frequently report the illusion of a presence. This presence—perceived as overwhelming or threatening—has been described as a generalized “monitoring entity” [3,16,17,18]. It may, thus, be inferred that the consistency of reports about a monitoring presence stems from a shared neurobiological basis, while its elaboration within cultural belief systems is likely a secondary process.

As suggested, experiences accompanying SP can be reliably characterized using a three-component model comprising: (1) intruder phenomena (a sensed presence, often perceived as threatening), (2) incubus phenomena (physical pressure on the chest and feelings of suffocation), and (3) unusual bodily experiences (including vestibular–motor hallucinations such as floating, flying, or out-of-body experiences). The first two components are closely interconnected and commonly resonate with culturally shaped narratives of numinous or malevolent encounters [19]. This model distinguishes between intruder and incubus phenomena (both of which resonate with cultural narratives of numinous experiences), and a third component referred to as “unusual bodily experiences.” Understanding this third component is particularly valuable in exploring neurophysiological mechanisms potentially shared by LSD- and 5-HT-mediated hallucinations, especially those involving altered spatial orientation of the self, e.g., out-of-body experiences (OBEs) and sensations of floating, levitation, or autoscopy (the experience of seeing one’s own body from an external perspective). Research suggests that the intruder experience—comprising perceived presence, intense fear, and visual/auditory hallucinations—stems from endogenous REM activation, which triggers a midbrain-mediated state of hypervigilance and attentional bias [20,21]. This mechanism allows the brain to interpret ambiguous stimuli in the context of environmentally relevant threats. In the absence of clear environmental cues, the sensation persists as a prolonged experience. This state influences how the brain shapes and integrates concurrent hallucinations, often imbuing them with supernatural or demonic qualities.

Further studies suggest that the intruder hallucination may arise from functional disturbances in the right superior parietal lobe (SPL), as well as disrupted multisensory bodily processing at the temporoparietal junction (TPJ) [22]. The same research group proposed that the projection of the intruder may stem from a genetically “encoded” body map in the right parietal region. These circuits are thought to influence aesthetic and sexual preferences related to body morphology. The hallucination may, thus, be caused by conflicting afferent and efferent neural signals, leading to atypical integration in regions responsible for the neural reconstruction of body representation. This hypothesis is supported by independent research [23], showing that focal electrical stimulation of the TPJ can induce an illusory sense of presence: a “shadow” figure mimicking the subject’s body position. Moreover, hyperactivity in the temporoparietal cortex in individuals with schizophrenia has been linked to erroneous attribution of self-generated actions to external agents [24].

**Figure 1 ijms-26-07233-f001:**
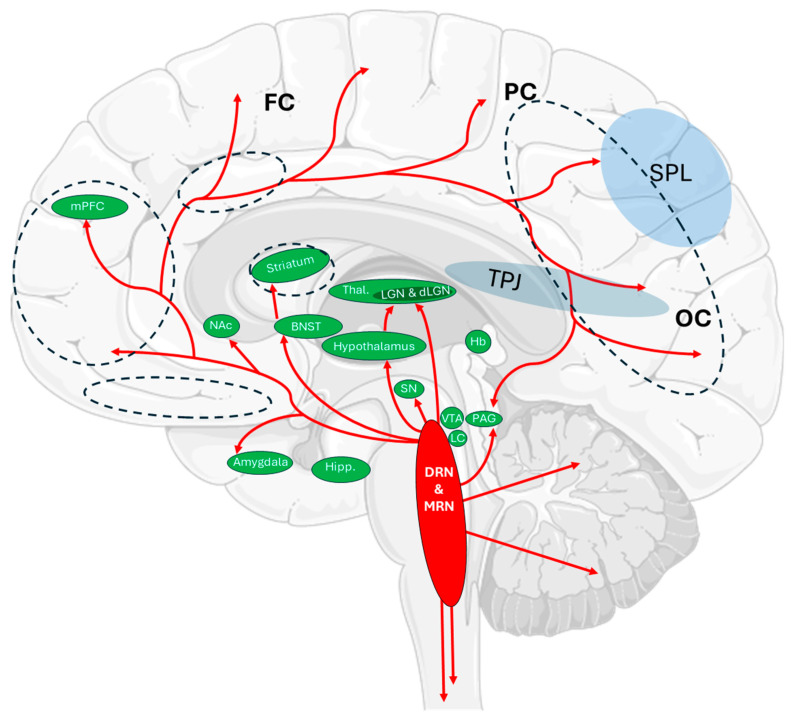
Schematic representation of serotonergic projections (red arrows) originating from the dorsal (DRN) and median (MRN) raphe nuclei. The dashed lines indicate brain regions with the highest density of 5-HT2A receptors. These projections modulate distinct neural functions through the topographically organized innervation of cortical and subcortical targets. mPFC, FC—cognitive flexibility and impulse control; NAc—regulation of social behavior and motivational salience of stimuli; amygdala, BNST, PAG, Hb—fear conditioning, anxiety expression, defensive responses; SPL, PC, TPJ—multisensory integration, spatial orientation, and body representation; LGN, dLGN, OC—visual signal transmission and perceptual shaping; SN, VTA—motor control and reward-based learning; LC—arousal and stress reactivity; hippocampus, thalamus, hypothalamus—memory encoding, sensory relay, circadian regulation and autonomic integration. Abbreviations: BNST—bed nucleus of the stria terminalis; dLGN—dorsal lateral geniculate nucleus; DRN—dorsal raphe nucleus; FC—frontal cortex; Hb—habenula; Hipp.—hippocampus; LC—locus coeruleus; LGN—lateral geniculate nucleus; mPFC—medial prefrontal cortex; MRN—median raphe nucleus; NAc—nucleus accumbens; OC—occipital cortex; PAG—periaqueductal gray; PC—parietal cortex; SN—substantia nigra; SPL—superior parietal lobe; Thal.—thalamus; TPJ—temporoparietal junction; VTA—ventral tegmental area. This figure was inspired by the serotoninergic connectivity schematic in Pourhamzeh et al. [25]. For detailed distribution maps of individual 5-HT receptor subtypes in the human brain, see [26,27]. Graphical elements from Servier Medical Art.

The second factor is the incubus phenomenon, characterized by chest pressure, breathing difficulties, shortness of breath, and pain: symptoms that are attributed to the impact of motor neuron hyperpolarization on the perception of respiration [28]. REM-induced muscle paralysis can create a subjective feeling of suffocation, particularly during attempts to take deep breaths, which may be interpreted as pressure on the chest or even a physical attack. These sensations closely align with traditional accounts of demonic assault. A shared theme in both types of experiences is the perception of an external aggressor, suggesting their interdependence.

The third factor, termed “unusual bodily experiences,” includes hallucinations involving disrupted spatial orientation of the self, e.g., OBE, sensations of floating, levitation, autoscopy, and often a concurrent feeling of bliss. These phenomena are believed to arise from a conflict between internal and external cues concerning body position, orientation, and movement.

The experience of a monitoring presence appears to uniquely distinguish SP from other atypical hallucinatory phenomena. Nevertheless, the “unusual bodily experiences” frequently accompanying SP are of particular interest due to potential similarities in neuronal activation patterns and molecular signaling pathways with those triggered by psychedelic substances.

The effects of LSD are characterized by profound alterations in bodily perception, synesthetic experiences, and disturbances in thought processes. The substance induces intense psychotropic effects often described as “mystical experiences”, which include altered states of consciousness, heightened introspection, a sense of unity with the environment, and distortions in the perception of time and space [14,29,30]. Other LSD-induced phenomena include metamorphic distortions of object and facial contours, the illusion of body boundary dissolution, and vivid, dynamic visual imagery from kaleidoscopic patterns to full evolving scenes. Notably, these experiences are associated with a deficit in sensorimotor gating, a mechanism potentially contributing to OBE [31].

Despite differences in the perceptual characteristics of hallucinations occurring during SP and LSD-induced states, the phenomenon of OBE—encompassing a broad spectrum of illusory experiences related to anomalous motion perception—may, in both cases, result from disrupted sensory integration and involve partially overlapping mechanisms.

At the molecular level, serotonergic neurons—which play a key role in regulating muscle tone and processing sensory information—are primarily located in the raphe nuclei but project widely throughout the brain to areas responsible for the integration and synchronization of this information, including the spinal cord, thalamus, and cerebral cortex [28,32,33,34]. As such, they are likely to contribute to the induction of movement-related hallucinations. These projections reach the cortex, limbic system, striatum, substantia nigra, and cerebellum via an extensive axonal network. The widespread distribution of 5-HT2A receptors [35] across various brain regions may significantly influence alterations in standard neuronal activity patterns due to their atypical activation. The intensity of these changes depends on both local neuronal interactions and broader network mechanisms. Enhanced functional connectivity among sensory cortical regions—mediated by 5-HT2A receptor activation—may facilitate the occurrence of synesthesia (a phenomenon in which stimulation of one sensory modality involuntarily triggers experiences in another, such as seeing sounds or tasting colors). It has been shown that, during REM sleep, neurons in the pons, occipital cortex, and lateral geniculate nucleus (LGN) fire in more intense bursts than during wakefulness [1]. Notably, the dorsal lateral geniculate nucleus (dLGN) of the thalamus has been identified as transmitting precise motion-related information to the visual cortex [36]. This higher-order thalamic nucleus relays a range of contextual signals that inform the visual cortex of changes in the visual scene independent of self-generated actions [36]. It can be hypothesized that increased LGN activity during REM sleep, along with excessive 5-HT2A receptor activation in the dLGN and visual cortex, may contribute to disturbances in motion perception. Thus, although perceptual hallucinations during SP and LSD-induced states differ in character, OBE—a phenomenon encompassing diverse illusory experiences related primarily to motion perception—may result from impaired sensory integration and rely on partially converging mechanisms. This does not suggest that SP- and LSD-induced states are phenomenologically or neurochemically identical. Rather, we propose that some shared disruptions—particularly in thalamo-cortical circuits integrating motion-related input—may lead to partially overlapping perceptual phenomena, such as OBE, across distinct states of consciousness.

### 2.2. Aberrations of Serotonin Signaling During Sleep Paralysis

Serotonin (5-HT, 5-hydroxytryptamine) functions as both a hormone and a central nervous system (CNS) neurotransmitter. Since it does not cross the blood–brain barrier, its synthesis from tryptophan occurs locally in serotonergic neurons, which are most densely concentrated in the brainstem raphe nuclei. Axonal projections of these neurons extend both rostrally (to the forebrain) and caudally (to the spinal cord) [37]. Serotonin is involved in the modulation of mood, motivation, memory, and cognitive function. In cooperation with acetylcholine, it regulates circadian rhythms, including the orexin-stimulated transition from REM sleep to wakefulness [6,38]. During REM sleep, raphe nucleus activity decreases, leading to reduced serotonin secretion and inhibition of the neurons responsible for its synthesis. Conversely, the activity of cholinergic neurons increases, promoting the production of acetylcholine, which is essential for REM initiation and maintenance. Increased 5-HT release accompanies the transition from REM to wakefulness. The activation of arousal systems and orexin-regulated stimulation of serotonergic neurons during persistent skeletal muscle atonia results in a dissociation of perceptual and motor systems [6]. This dissociation is accompanied by intense fear, activation of fear circuits, and a sympathetic nervous system response that further intensifies 5-HT release [6]. Non-physiological aberrations in serotonin levels disrupt neuronal signaling and interfere with the accurate recognition and processing of both external and internal stimuli.

### 2.3. 5-HT Receptors

The site of serotonin binding is the 5-HT receptor family. Seven distinct families of serotonin receptors have been identified [39,40,41,42,43,44]. Most of these belong to the G protein-coupled receptor (GPCR) superfamily [45,46,47,48,49], with the exception of the 5-HT3 family, which comprises ligand-gated ion channels. 5-HT receptors are involved in both excitatory and inhibitory neurotransmission. Different serotonin receptor families show specific preferences for coupling with distinct G proteins, a feature that determines their functional diversity. The 5-HT2A receptor (5-hydroxytryptamine receptor 2A, or 5-HT2Ar) belongs to the 5-HT2 subfamily, which also includes 5-HT2B and 5-HT2C receptors. These are grouped together due to their high degree of structural homology.

Notably, 5-HT2A receptors are the most widely expressed among all serotonin receptor subtypes and are found in nearly every mammalian tissue examined to date. In the central nervous system, they are particularly abundant in the neocortex, where they are organized according to the cytoarchitectonic banded structure, predominantly located in layer V of pyramidal neurons [12,50,51,52]. Upon activation, these receptors can regulate perception, cognitive and mood alterations, and neurogenesis [53,54]. Studies involving the deactivation of 5-HT2A receptors have shown that they are essential for the promotion of hallucinogenesis, as blocking them prevents the characteristic perceptual effects of psychedelics such as LSD or psilocybin [55,56]. Thus, in contrast to the multitude of other serotonin receptors, 5-HT2Ar possesses a unique potential for generating sensory hallucinations and OBEs, especially in the context of SP.

### 2.4. The Role of the 5-HT2A Receptor in Physiological Sensory Processing

Historically, the 5-HT2A receptor has been primarily studied in the context of its role in mediating the effects of psychedelic compounds. However, the development of modern research techniques and the growing interest in understanding its function in sensory processing—which encompasses dynamic encoding and modulation of sensory inputs—offer a broader perspective on its physiological relevance. Recent studies suggest that the 5-HT2A receptor may act as a modulator of sensory gain under baseline non-pharmacological conditions. Evidence is particularly abundant for visual systems due to their experimental accessibility and the high spatial and temporal precision of available measurement techniques. Anatomical studies have demonstrated high 5-HT2A receptor expression in somatomotor and auditory cortices [57], while functional data from studies using psychedelics support its role in modulating tactile and auditory perception [58]. Currently, most insights into the receptor’s contribution to sensory processing are derived from pharmacological interventions, while direct evidence of its function under physiological conditions remains limited. Notably, no studies to date have examined 5-HT2A-dependent sensory processing in humans under naturalistic conditions, i.e., without receptor ligands.

Recent optogenetic studies have provided direct evidence of the receptor’s role in physiologically modulating neuronal responses in the mouse visual cortex, making it the most thoroughly characterized model to date for linking molecular activation with perceptual output. Barzan et al. [59] used optogenetics to selectively activate endogenous 5-HT2A receptors in specific neuronal populations of the mouse visual cortex: pyramidal cells and parvalbumin-positive (PV) interneurons. Activation of 5 HT2A receptors in pyramidal neurons led to increases in both spontaneous and stimulus-evoked neuronal activity (by approximately +48% and +70%, respectively). In contrast, activation in PV interneurons resulted in excitation of the interneurons themselves, accompanied by strong inhibition of neighboring pyramidal cells, which produced a notable decrease in visual responsiveness (around −35%). Notably, simultaneous activation of both neuronal types (PV interneurons and pyramidal cells)—which best reflects endogenous serotonin signaling—selectively suppressed stimulus-driven responses without increasing baseline activity, suggesting a precise gain regulation mechanism. Furthermore, PV activation reduced the power of local field potential (LFP) oscillations, linking 5-HT2A receptor activity to network rhythmicity. A similar effect was observed in studies on macaques, where the microiontophoretic application of serotonin in the primary visual cortex (V1) led to a reduction in visually evoked neuronal responses, an effect that was abolished by inhibition of 5-HT2A receptor. This finding confirms the critical role of 5-HT2A receptors in gain modulation also in primates [60,61].

Those findings confirm that 5-HT2A receptors function as dynamic regulators of sensory signal strength, demonstrate that their effects depend on coordinated activation of excitatory and inhibitory neurons, and reveal a mechanistic bridge between molecular signaling and cortical network activity. Most importantly, they provide one of the first direct demonstrations that 5-HT2A receptors modulate sensory response strength and selectivity under physiological conditions through cell-type-specific control of excitability and network oscillations. As such, these results offer a critical foundation for understanding how endogenous serotonergic activity shapes perception and how SP or psychedelic drugs may alter sensory processing.

### 2.5. The Role of 5-HT2A in Shaping Perception: Molecular Pathways and Cortical Rhythms

The functional consequences of 5-HT2A receptor activation extend from molecular changes in individual neurons to the reorganization of activity across entire cortical networks (Figure 1). At the cellular level, the Gq/11 pathway leads to the opening of non-selective cation channels, membrane depolarization, and the suppression of calcium-dependent afterhyperpolarization (sAHP), thereby increasing the excitability of pyramidal neurons and enhancing their cellular gain [62,63]. The variability of this response—determined by membrane potential, excitation/inhibition balance, and local network activity—may account for individual susceptibility to hallucinations, particularly under conditions of reduced external sensory input, such as in sleep paralysis. These same neurons frequently co-express 5-HT1A receptors, which, through the activation of K^+^ channels, induce hyperpolarization, an effect opposite to that mediated by 5-HT2A [64]. Depending on the initial membrane potential, serotonergic effects may display a bipolar pattern—initially inhibitory (via 5-HT1A) and subsequently excitatory (via 5-HT2A)—providing a potential temporal switching mechanism for neuronal responsiveness [65,66]. In parallel, serotonin activates PV^+^ interneurons, leading to the inhibition of neighboring pyramidal cells and selective signal filtering [67]. In vitro, 5-HT2A activation increases spontaneous EPSCs in layer V pyramidal neurons independently of presynaptic activity in a manner dependent on the presence of Ca^2+^/Sr^2+^ ions and the action of synaptotagmin III, indicating asynchronous glutamate release [11].

At the network level, differential expression of 5-HT2A receptors enables local control over excitation/inhibition (E/I) balance and circuit responsiveness. As previously shown, their activation—producing either excitation or inhibition depending on cellular localization—modulates both neuronal excitability and network rhythms [59].

In humans, EEG studies have shown that psilocybin, a potent 5-HT2A agonist, reduces alpha power in parietal–occipital regions and suppresses N170 visually evoked potentials, a negative deflection in the local field potential occurring approximately 170 ms after a visual stimulus, typically associated with face perception. Both effects were blocked by ketanserin, confirming 5-HT2A dependence [53]. Additionally, psilocybin and LSD reduce the signal-to-noise ratio, induce cortical desynchronization, and disrupt stimulus-specific response selectivity—particularly for facial and tactile inputs—which has been interpreted as a physiological mechanism underlying reduced perceptual resolution and increased attribution of significance to endogenous stimuli in hallucinatory states [54,68,69]. Concurrently, increased gamma synchrony and unstable slow-wave oscillations have been observed, suggesting a complex frequency-specific reorganization of network dynamics [69].

An increasing body of evidence indicates that the alternative β arrestin 2-dependent signaling pathway—engaging MAP kinases (ERK1/2), receptor recycling, and the expression of ion channels—plays an independent functional role in modulating neuronal reactivity [70,71,72,73]. Although the direct impact of this pathway on rhythmic parameters—such as power, coherence, or frequency-specific responsiveness—has not yet been fully characterized, the available data support a hypothesis that, by regulating local E/I balance and cellular excitability, β arrestin 2 signaling may indirectly shape oscillatory dynamics. This remains one of the key unresolved questions, discussed further in the section on outstanding issues. In conditions of limited external sensory input—such as during sleep paralysis—such modulation may facilitate the transient disorganization of network activity and the emergence of hallucinatory endogenous representations.

### 2.6. 5-HT2A Receptor Pathophysiology and Its Hallucinogenic Potential

Dysfunction and/or altered distribution of the 5-HT2A receptor contributes to the pathogenesis of various neurological disorders. A reduced density of 5-HT2A receptors within the cell membrane has been implicated as a predisposing factor in the onset of schizophrenia [74]. The importance of this receptor is further underscored by the fact that many antipsychotic medications prescribed for patients with depression and/or anxiety disorders—particularly selective serotonin reuptake inhibitors (SSRIs)—target this receptor.

The ability of specific 5-HT2Ar ligands to generate receptor-specific intracellular signaling profiles, resulting in divergent effects, is the subject of ongoing investigation. While hallucinogens such as LSD and psilocybin bind to this receptor with high affinity and are frequently associated with recreational use, other structurally related compounds—such as lisuride and ergotamine—are used therapeutically in the treatment of migraines and Parkinson’s disease and notably do not induce hallucinations [75,76,77,78].

Under physiological conditions of serotonin secretion, the activation of 5-HT2A-receptor-dependent signaling pathways does not lead to hallucinogenic effects comparable to those elicited by psychedelics [79]. However, there are reports of visual and auditory hallucinations [80] occurring during serotonin syndrome, a condition triggered by excessive therapeutic doses of SSRIs such as sertraline, a potent selective serotonin reuptake inhibitor. Based on available evidence, it can be hypothesized that the 5-HT2A receptor not only plays a central role in the emergence of sensory disturbances observed during sleep paralysis, but that the modulation of 5-HT2A-dependent pathways may also be significant in the treatment of complex neuropsychiatric disorders.

### 2.7. 5-HT2A-Receptor-Dependent Molecular Signaling Pathways Promoting Hallucinogenic Effects

Although both serotonin and psychedelic compounds bind to 5-HT2A receptors, the downstream effects of this binding can vary significantly. This variability arises from several factors, including differences in receptor-associated or independent protein signaling [42,43,81,82], receptor localization in the polarized neuronal membrane, and competition from endogenous ligands that influence selection of specific intracellular pathways, leading to receptor internalization [83]. A key element of this complexity is the phenomenon of biased agonism, a mechanism by which different ligands binding to the same receptor can selectively activate distinct intracellular signaling cascades [84,85,86]. As a result, the precise molecular basis of hallucinations occurring during sleep paralysis remains complex and, hitherto, not fully elucidated.

According to the model of the aforementioned “biased agonism” [84,85,86], agonist ligands possess the intrinsic ability to stabilize different conformational states of the receptor and, consequently, show differential propensity to activate specific signaling cascades associated with receptor-coupled proteins [13,87]. Moreover, agonists display varying affinities for subsets of GPCR conformational states, which, in turn, selectively activate distinct intracellular signaling pathways. Additionally, an agonist may activate certain pathways while inhibiting others, explaining why different substances targeting 5-HT2A receptors elicit distinct cellular responses. This concept has recently gained scientific attention due to the hypothetical potential to develop drugs that activate only therapeutically beneficial intracellular signaling patterns while avoiding undesirable side effects [84,88].

A ligand’s selective affinity for specific GPCR states determines its preference for activating distinct signaling cascades, ultimately leading to diverse transcriptomic profiles depending on the pathway engaged. A hallmark of 5-HT2A receptors is their preferential coupling to Gq/11 proteins (Figure 2A), whereas, for example, 5-HT1 receptors predominantly activate Gi/o proteins, which are pertussis-toxin-sensitive [43,44,88,89,90]. The activation of specific G proteins initiates distinct secondary messenger cascades. The primary signaling route for 5-HT2A receptors following 5-HT or agonist binding is the Gq/11-mediated activation of phospholipase C (PLC), leading to inositol 1,4,5-trisphosphate (IP_3_) production (the PLC-IP_3_ pathway) and subsequent Ca^2+^ release from the endoplasmic reticulum into the cytoplasm [9]. This increase in intracellular Ca^2+^ activates calmodulin and calcium/calmodulin-dependent kinases (CaMKs) and protein kinase C (PKC), which, in turn, can stimulate the MAPK/ERK pathway, a critical regulator of gene transcription and synaptic plasticity (Figure 2A—blue pathway).

Although the PLC-IP_3_ cascade is the most thoroughly characterized signaling pathway downstream of the 5-HT2A receptor, it may neither be the only nor the most functionally relevant mechanism contributing to hallucinogenic effects. In addition to G protein-mediated mechanisms, β-arrestin (βarr)-dependent pathways seem to play a pivotal role in intracellular processes triggered by 5-HT2A receptor activation (Figure 2A—red pathway). βarrs promote receptor internalization, thereby terminating the signal [91]. However, they also act as scaffolds facilitating signal transduction from GPCRs to intracellular effectors, functioning in opposition to internalization [91,92]. Two isoforms are distinguished: β-arrestin-1 (βarr1) and β-arrestin-2 (βarr2). While βarrs may not influence the hallucinogenic effects of DOI [93] and play a secondary role in LSD-induced hallucinations, they might be key mediators of hallucinogenesis during SP. The following sections discuss the role of βarrs, with particular focus on their relevance to serotonergic transmission via 5-HT2A receptors.

A comparative pharmacological profile of 5-HT2A receptor agonists is presented in Table 1.

### 2.8. Serotonin

Although 5-HT is not considered hallucinogenic, research [94] shows that its signaling via βarr2 in the prefrontal cortex differs from that of psychoactive N-methyltryptamines. Both 5-HT and N-methyltryptamines induce head-twitch responses and activate cortical signaling but through distinct mechanisms. 5-HT forms a complex with βarr2, Src, and Akt, and disruption of any aforementioned components prevents full expression of HTR, highlighting the importance of this complex [94].

Notably, the same research group demonstrated that, while N-methyltryptamines act through 5-HT2A receptors to induce HTR, the mechanism is βarr2- and Akt-independent. Other studies [95] suggest βarr2 modulates 5-HT-dependent responses (Figure 2A). It was reported that serotonin-induced ERK1/2 activation via 5-HT2A receptors depends on β-arrs, whereas DOI-induced activation does not require them [93]. 5-HT-induced HTR depends on βarr2, while DOI-induced responses are βarr2-independent. Moreover, the 5-HT2A receptor itself is essential, as HTR is absent in HTR2A knockout mice [9,93]. Interestingly, clozapine, an atypical antipsychotic and 5-HT2A receptor antagonist, may compensate for hyperserotonergic responses (Figure 2A).

### 2.9. LSD

While 5-HT2A receptors preferentially couple to Gq/11 proteins, downstream signaling may vary depending on the bound ligand. LSD has been shown to activate the PLC-IP_3_ pathway with low efficacy (Figure 2B—grey pathway), but behavioral activity does not always correlate with in vitro or in cellulo efficacy [39,43,78,88,96,97]. LSD has also been shown to activate the phospholipase A2–arachidonic acid (PLA2–AA) pathway in heterologous expression systems (Figure 2B—orange pathway) [43,87,98]. Here, PLA2 is activated through Gq/11 signaling, and AA serves as a precursor for lipid mediators essential in intra- and intercellular communication. AA and its metabolites modulate ion channel permeability, leading to depolarization of pyramidal neurons. This depolarization activates subsequent neurons in various brain regions, resulting in sensory integration and perceptual changes (similar processes are discussed below regarding psilocybin). This depolarization activates subsequent neurons in various brain regions, resulting in sensory integration and perceptual changes [14].

LSD has also been shown to activate Gi/o-dependent pathways in heterologous expression systems and murine models [98]. The 5-HT2A-receptor-dependent signaling pathways discussed so far provide only a limited perspective on the complexity of interactions mediated by this serotonin receptor. This article has placed particular emphasis on the role of βarrs in mediating the hallucinogenic effects of each of the substances addressed.

Studies indicate that LSD’s psychedelic activity requires βarr2, but not βarr1 [99]. In mice, LSD stimulates motor activity in wild-type (WT) and βarr1 knockout (βarr1-KO) mice but not in βarr2 knockout (βarr2-KO) mice. In both WT and βarr1-KO mice, LSD robustly induces a range of behaviors characteristic of psychedelic drug action, including head-twitching (the head-twitch response, HTR, observed in rodent studies, is a widely used behavioral marker of human psychedelic effects), a shortened or non-sequential grooming sequence and/or grooming restricted to a single body region (mouse grooming behavior serves as a behavioral indicator of altered tactile perception due to its typically stereotyped chain organization: face, flanks, hind limbs, tail), backward locomotion, and nose-poking (a behavioral marker of altered exploratory behavior; changes in nose-poking reflect disruptions in perception, attention, or sensory arousal). In contrast, βarr2-KO mice display minimal head-twitch responses, and LSD does not affect other such behaviors (e.g., grooming sequences remain complete and are rarely disrupted) [99].

HTR is a reliable behavioral marker for 5-HT2A-receptor-agonist-induced psychedelic activity as it is absent with non-hallucinogenic agonists like lisuride and ergotamine [100]. The antagonist MDL100907 (MDL) [101], which blocks LSD’s actions at 5-HT2A receptors, also normalizes LSD-disrupted prepulse inhibition (PPI) in WT and βarr1-KO mice but not in βarr2-KO mice. In βarr1-KO mice, haloperidol is required to restore PPI. Together, these findings underscore βarr2′s critical role in LSD-induced hallucinogenic responses and reinforce 5-HT2A receptor’s role in mediating these effects [99] (Figure 2B).

It is worth noting that findings from independent research [102] suggest that βarr-biased 5-HT2A agonists induce rapid tolerance upon repeated administration and may exhibit behavioral profiles akin to antipsychotic agents.

### 2.10. Psilocybin

As with LSD, psilocybin interacts not only with serotonergic systems, but also with other receptor systems, including dopaminergic receptors, complicating the elucidation of its exact physiological effects. Nevertheless, studies suggest that psilocybin’s psychotomimetic effects (i.e., its ability to induce temporary states resembling psychosis, including hallucinations and delusions) can be blocked by ketanserin (a 5-HT2A receptor antagonist) or risperidone (an atypical antipsychotic), while haloperidol (a typical dopamine antagonist) enhances these effects [103]. This implies that psilocybin-induced psychosis is predominantly driven by 5-HT2A receptor activation, independent of dopaminergic influence.

5-HT2A receptor activation can disrupt normal neuronal function, causing sensory distortions and altered consciousness. Psilocybin-induced visual hallucinations, along with the modulation of alpha oscillations and N170 visually evoked potentials, have been shown to be mediated by 5-HT2A receptors [53]. These findings may extend beyond drug-induced states to neuropsychiatric hallucinations. Alpha oscillations (8–12 Hz) regulate excitability in sensory cortical networks through inhibition [104,105,106] and play a critical role in maintaining excitability balance in the visual cortex. Their suppression following 5-HT2A receptor activation may destabilize the balance between externally and internally driven neural activity, potentially shifting perception toward internally generated experiences, which are critical for the emergence of visual hallucinations.

N170 potential (a negative deflection in a local field potential of ~170 ms) is essential for perceiving coherent and meaningful visual stimuli. Their attenuation by 5-HT2A receptor activation may relate to visual distortions [53]. These electrophysiological effects, previously linked to 5-HT_2_A activation, are attenuated by ketanserin, confirming the receptor’s role in sensory network excitability. Psilocybin-induced functional neuronal changes may be explained by 5-HT2A-receptor-dependent intracellular calcium mobilization via activation of the PLC-IP_3_ molecular pathway [107].

### 2.11. Unresolved Questions and Working Hypothesis

Despite the growing body of evidence on 5-HT2A-dependent network phenomena, a key question remains unresolved: how could the relatively slow intracellular effects of β arrestin 2-dependent signaling—such as gene transcription or cytoskeletal reorganization, typically occurring over minutes to hours—account for the rapid onset of hallucinatory episodes during SP, which often last only seconds to a few minutes?

To address this temporal discrepancy, we propose an intermediary mechanism whereby β arrestin 2 may recruit non-canonical effectors such as ERK1/2 kinases within tens of seconds following receptor activation [70,71,72,73]. Once activated, the ERK cascade can rapidly modulate local E/I balance by phosphorylating AMPA receptors (e.g., GluA1) and potassium channels (e.g., Kv/Kir), leading to increased signal amplification in pyramidal neurons, while simultaneously reducing GABA_A receptor clustering on PV interneurons, thereby weakening fast inhibitory control [108,109].

Even minor shifts in the E/I balance may destabilize network rhythmicity: both in vivo studies and computational models demonstrate that a mere 10–20% reduction in PV-mediated inhibition can disrupt alpha coherence, trigger bursts of gamma activity, and produce broadband desynchronization [110,111,112]. Under SP conditions—characterized by suppressed thalamo-cortical sensory input and internally driven cortical dynamics—such β arrestin 2/ERK-dependent E/I shifts may promote the transient disinhibition of top-down imagery, resulting in the emergence of vivid endogenous representations [21,113,114].

**We therefore propose that this signaling pathway does not directly induce hallucinations, but rather creates a state of network susceptibility that facilitates their abrupt emergence in sensory areas.** This hypothesis is empirically testable: selective inhibition of ERK1/2 in layer V pyramidal neurons during REM-like states should attenuate or abolish visual hallucinations associated with SP without affecting muscular atonia. As such, the model offers a coherent framework for linking slow molecular signaling with fast perceptual distortions, providing novel avenues for both experimental research and therapeutic interventions.

## 3. Summary

The above findings suggest that, under physiological conditions, 5-HT primarily activates the PLC-IP_3_ signaling cascade via 5-HT2A receptors. However, during SP, neurotransmitter imbalance may shift signaling toward βarr2-dependent intracellular pathways. This altered signaling may manifest as visual and auditory hallucinations. Such insights pave the way for novel therapeutic strategies targeting serotonergic hallucinations, whether as side effects of SSRI use or in other 5-HT2A-receptor-linked dysfunctions. These findings open promising directions for future research, warranting further experimental investigation. **We propose a mechanistic hypothesis whereby β-arrestin-2-mediated ERK activation transiently disrupts E/I balance and oscillatory stability, bridging the temporal gap between molecular signaling and the rapid onset of hallucinatory perception in SP.**

## 4. Methodology

The current scoping review was conducted to synthesize and conceptualize existing molecular data potentially linking the 5-HT2A receptor and β-arrestin-2 with hallucinatory phenomena occurring during sleep paralysis. This review closely observed the PRISMA-ScR guidelines.

A comprehensive search strategy was employed using major scientific databases, including PubMed, ScienceDirect, Springer Nature Link, Science Signaling, Neuropsychopharmacology, and ResearchGate. No publication year restrictions were applied. The only exclusion criterion was access-related: sources were included only if the whole text was available through online academic databases. No physical libraries or closed-access repositories were included in the search protocol. The search aimed to capture the most relevant and up-to-date knowledge on 5-HT2A receptor signaling pathways, β-arrestin-2 molecular functions, and their potential interplay in neuropsychiatric and hallucinatory processes.

The following key words were used in the primary screening: hallucinations, serotonin, 5-HT, sleep paralysis, 5-HT2A (receptor), β-arrestin.

The initial search (OR’ criteria) yielded over 10,000 articles. Within those articles, titles and abstracts were further screened down for relevance to 250 fully read articles. When available, full texts were retrieved with DOI/PMID from original journal platforms. In general, only full-text sources were included in the final synthesis. However, an exception was made for selected abstracts, which provided valuable and most-up-to-date information, when no full-text article on the topic existed. A total of 114 sources were included in the final review.

Inclusion criteria were based on conceptual relevance and specificity to the topic. Articles were included if they focused specifically on molecular mechanisms involving the 5-HT2A receptor and β-arrestin-2. When possible, primary research articles were prioritized over reviews. Negative selection criteria excluded non-specific studies, such as those discussing broader interactions of ligands with multiple serotonin receptor subtypes (e.g., 5-HT2A/C) or those addressing β-arrestins in general without discussing β-arrestin-2 explicitly.

No formal protocol was registered prior to conducting the review. While no automated charting tool was used, all included sources were reviewed in full and critically analyzed. Appraisal of sources focused on scientific reliability, thematic relevance, mechanistic clarity, and contribution to a coherent conceptual model.

Special attention was given to the accuracy, credibility, and thematic coherence of the data. The selection process emphasized both specificity and integrative scope to ensure a comprehensive understanding of the topic without omitting any essential findings.

Generative artificial intelligence (GenAI) was not implemented in the screening process nor in data selection, writing, or figure generation.

## Figures and Tables

**Figure 2 ijms-26-07233-f002:**
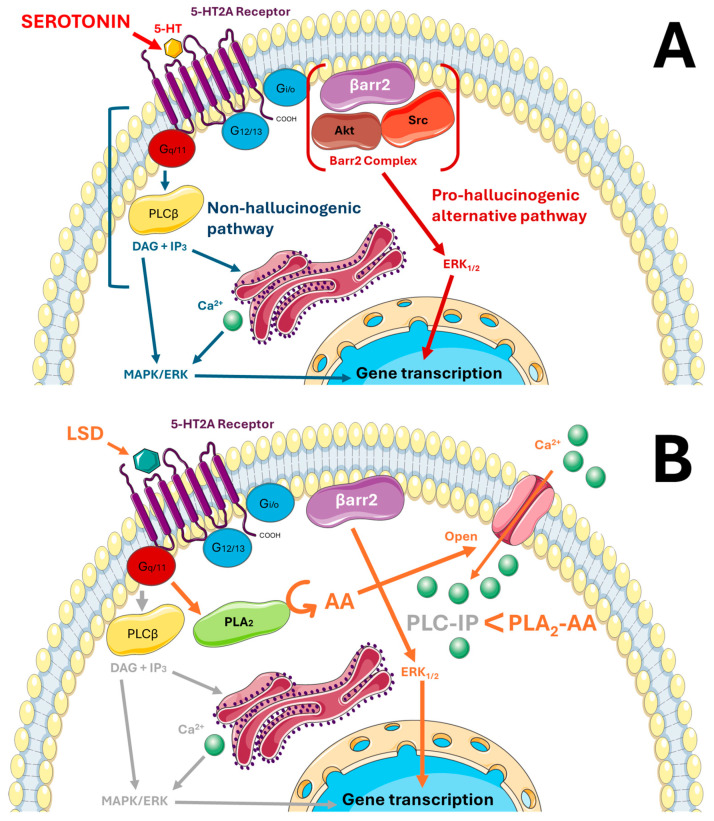
(**A**) Comparison of the preferentially activated non-hallucinogenic PLC–IP_3_ signaling pathway (blue) and the proposed molecular cascade promoting SP-related hallucinations via β-arrestin-2 (red). The β-arrestin-2-dependent pathway involves the 5-HT-induced atypical activation of the 5-HT2A receptor, leading to the formation of a βarr2–Src–Akt signaling complex. The aforementioned complex triggers ERK1/2 phosphorylation and further gene transcription. (**B**) Intracellular signaling pathways activated by LSD via the 5-HT2A receptor. LSD activates the PLC-IP_3_ signaling pathway with moderate efficacy (grey), leading to MAPK/ERK activation, but preferentially engages the PLA_2_–arachidonic acid (AA) pathway (orange). Activation of the PLA_2_–AA pathway leads to calcium influx and neuronal depolarization through ion channel opening. The presence of β-arrestin-2 is required for the hallucinogenic effect. Self-compiled in power-point; graphical elements adapted from Servier Medical Art.

**Table 1 ijms-26-07233-t001:** Comparative pharmacological profile of 5-HT_2_A receptor agonists.

Compound (Common Name)	Receptor Interactions	Activated 5-HT_2_AR-Dependent Intracellular Pathways	Potential Behavioral Effects
5-hydroxytryptamine—5-HT (serotonin)	5-HT_1_–5-HT_7_	PLC-IP_3_ (predominantly activated), βarr2	No hallucinations (via PLC-IP_3_)/potential hallucinations (via βarr2)
lysergic acid diethylamide (LSD)	5-HT_2_A, 5-HT_1_A, 5-HT_2_C, 5-HT_1_B, 5-HT_1_D, 5-HT_5_A, 5-HT_6_, 5-HT_7_, D_1_, D_2_, D_4_, α_2_A, α_2_B, α_2_C, α_1_A, α_1_B (predominantly 5-HT_2_A; additional binding sites reported)	PLA2-AA (preferentially engaged by LSD over PLC-IP_3_), PLC-IP_3_, βarr2	Hallucinations
2,5-dimethoxy-4-iodoamphetamine (DOI)	5-HT_2_A, 5-HT_2_C, 5-HT_2_B (predominantly 5-HT_2_A; additional binding sites reported)	PLC-IP_3_ (predominantly activated)	Hallucinations
4-phosphoryloxy-N,N-dimethyltryptamine—4-PO-DMT (psilocybin)	5-HT_2_A, 5-HT_1_A, 5-HT_2_C, 5-HT_1_B, 5-HT_7_ (predominantly 5-HT_2_A; additional binding sites reported)	PLC-IP_3_ (predominantly activated)	Hallucinations

## Data Availability

Not applicable, as no new data were created.

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
