# Peer review of "Molecular Pathways Potentially Involved in Hallucinatory Experiences During Sleep Paralysis: The Emerging Role of β-Arrestin-2"

_ijms, 2025, doi:10.3390/ijms26157233_

Round 1

Reviewer 1 Report

Comments and Suggestions for Authors

This manuscript is devoted to an interesting and poorly studied topic, such as the mechanisms of hallucinations during sleep paralysis. The authors thoroughly searched for related literature and tried to create a concept of the prevalence of serotonergic signaling abnormalities during the phenomenon.  However, some questions remained unclear from the read manuscript.

  1. The authors suggested that hallucinatory experience during sleep paralysis and hallucinations induced by different psychoactive drugs (LSD or psilocybin) are the same. However, I could not assume this from the manuscript. Most discussion on this point seems to be very speculative. Is there any firm evidence of this point of view?
  2. The authors referred actively the studies by Baland Jalal, but did not refer to his good review on the same theme, specifically Jalal, B. The neuropharmacology of sleep paralysis hallucinations: serotonin 2A activation and a novel therapeutic drug. Psychopharmacology 235, 3083–3091 (2018). https://doi.org/10.1007/s00213-018-5042-1. How does their review differ from the article by Jalal ?  What is new?
  3. Probably, some limitations in data interpretation should be indicated in the manuscript.

Author Response

Answers to the comments of the Reviewer 1

First of all we would like to thank the Reviewer for the valuable comments, that enhanced the clarity of our manuscript.

This manuscript is devoted to an interesting and poorly studied topic, such as the mechanisms of hallucinations during sleep paralysis. The authors thoroughly searched for related literature and tried to create a concept of the prevalence of serotonergic signaling abnormalities during the phenomenon.  However, some questions remained unclear from the read manuscript.

All inclusions, corrections or clarifications as well as new citations are highlighted in yellow.

  1. The authors suggested that hallucinatory experience during sleep paralysis and hallucinations induced by different psychoactive drugs (LSD or psilocybin) are the same. However, I could not assume this from the manuscript. Most discussion on this point seems to be very speculative. Is there any firm evidence of this point of view?

Further explanation and clarification along with new sources were provided in the manuscript.

  1. The authors referred actively the studies by Baland Jalal, but did not refer to his good review on the same theme, specifically Jalal, B. The neuropharmacology of sleep paralysis hallucinations: serotonin 2A activation and a novel therapeutic drug. Psychopharmacology 235, 3083–3091 (2018). https://doi.org/10.1007/s00213-018-5042-1. How does their review differ from the article by Jalal?  What is new?

In the citation of Jalal, 2018, the title was mistakenly changed to his 2017 paper. The reference 6 was corrected in the literature. The new element of current review is the working hypothesis presented in the chapters: “Unresolved questions and working hypothesis” and Summary, which states, that rather than directly causing hallucinations, the βarr2 pathway transiently disrupts E/I balance and oscillatory stability in the neurons which induces a state of sensory network susceptibility.

  1. Probably, some limitations in data interpretation should be indicated in the manuscript.

The whole new subchapter “Unresolved questions and working hypothesis” was included in the manuscript, where the unknowns are briefly discussed and a new working hypothesis is included alongside a potential model to verify its validity. Appropriate change was also included in the abstract of the manuscript. Furthermore, two new subchapters on the physiological role of 5-HT2A receptor in sensory processing and perception shaping were added. Numerous smaller changes were also included throughout the manuscript.

Reviewer 2 Report

Comments and Suggestions for Authors

The manuscript provides an interesting overview of the serotonergic mechanisms, specifically 5-HTâ‚‚A receptor pathways, underlying hallucinatory experiences during sleep paralysis. While the review brings forward a compelling topic at the intersection of molecular neuroscience and consciousness studies, several areas of improvement will enhance the manuscript’s clarity and accessibility to a broad readership.

Major comments:

  • Figures 1 and 2 are conceptually and visually redundant. I suggest merging them into a single, consolidated schematic (e.g., Figure 1A and 1B or a unified pathway diagram) that distinguishes the LSD versus 5-HT pathways using color-coded arrows (e.g., blue for LSD, orange for serotonin).
  • The pyramidal neuron illustration in these figures is misleading and should be removed. The 5-HTâ‚‚A receptor is not exclusive to pyramidal cells; rather, it is broadly expressed across multiple neuronal and even non-neuronal cell types throughout the brain and peripheral organs, as acknowledged by the authors themselves.
  • I strongly recommend adding a systems neuroscience-style figure that illustrates serotonergic projections from the raphe nuclei and the distribution of 5-HT receptor subtypes across different brain regions. Such a figure would help bridge intracellular signaling pathways with large-scale neural circuit dynamics, particularly relevant to the discussion starting at line 167.
  • The role of 5-HTâ‚‚A receptors in sensory processing under normal physiological conditions remains underexplored. A brief summary of what is known from animal models or human studies would contextualize their dysfunction in SP and psychedelic states.
  • The manuscript’s goal—bridging molecular signaling with cognitive and perceptual states—would benefit from a dedicated paragraph linking molecular and circuit-level mechanisms. Consider expanding the discussion to include electrophysiological findings from animal and human studies. For instance, going beyond the brief mention of alpha oscillations and the N170 potential (line 362) would enrich the discussion, especially regarding network-level correlates of hallucinations.
  • In a review, a section dedicated to “outstanding questions and unresolved issues” is welcomed. Please add a short paragraph highlighting what the authors think are the most pressing unknowns.
  • The Summary section would benefit from a more integrative perspective. Specifically:
    • Emphasize the temporal mismatch between molecular signaling (e.g., β-arrestin-2 pathway activation) and the fast, transient nature of hallucinations during SP.
    • Discuss how the electrical state of the 5HT2A-expressing neurons (e.g., depolarization thresholds, excitatory/inhibitory balance) might modulate susceptibility to hallucinations.

Minor comments:

  • Lines 49–50: Please add a reference supporting the claim that altered serotonergic signaling primarily disturbs pyramidal neuron function.
  • Line 85: the authors state “… not identical to those after ingesting psychedelic substances, some important similarities do exist” but fail to describe which ones. Please specify the nature of these similarities (e.g., visual distortions, altered sense of self or time, etc.).
  • Line 94: The sentence on hypnagogic and hypnopompic phases adds little without further elaboration. Removing this sentence will improve clarity, but if retained, it should be better integrated into the paragraph and define both terms for clarity.
  • Line 98: “In specific regions…” > please name these brain regions explicitly. Ideally, they would be visualized in the proposed new figure.
  • Line 109: Clarify the “three-component model” of SP. If referring to intruder, incubus, and unusual bodily experiences, please list them explicitly and define as needed.
  • Line 115: Provide a brief definition of autoscopy at first mention.
  • Lines 123 onward: Numerous brain areas are cited (e.g., SPL, TPJ, right parietal cortex). It would be valuable if these regions were annotated in the suggested systems-level figure.
  • Line 168: Add a citation to substantiate the claim that these regions “play a key role in regulating muscle tone and processing sensory information.”
  • Line 177: Define synesthesia briefly.
  • Line 208: Add reference (perhaps reference #6) for the mechanism discussed.
  • Line 224 & 226: Add appropriate citations for claims regarding the role of these substances in cognitive/mood alterations and their essential role in hallucinogenesis.
  • Lines 254 onward: The introductory paragraph of this section is a dense, single sentence that is difficult to follow. Please rephrase for clarity and introduce the concept of biased agonism with a brief explanation at first use.
  • Lines 272 onward: Specify which figure is being referenced (presumably Figure 2) and ensure figures are cited appropriately throughout.
  • Table 1: Please adopt a consistent naming convention for the column “compound”, such as Chemical name (common name) or vice versa.
  • Line 317: The phrase “in this review…” is misplaced, as it occurs near the end of the article. Consider removing or rephrasing.
  • Line 355: Define psychotomimetic.
  • Line 362: Define N170 as the negative deflection in local field potential ~170 ms after a visual stimulus.

Author Response

Answers to the comments of the Reviewer 2

First of all, we would like to thank the Reviewer for the comprehensive review and valuable comments, that greatly enhanced the clarity of our manuscript.

The manuscript provides an interesting overview of the serotonergic mechanisms, specifically 5-HTâ‚‚A receptor pathways, underlying hallucinatory experiences during sleep paralysis. While the review brings forward a compelling topic at the intersection of molecular neuroscience and consciousness studies, several areas of improvement will enhance the manuscript’s clarity and accessibility to a broad readership.

Major comments:

  • Figures 1 and 2 are conceptually and visually redundant. I suggest merging them into a single, consolidated schematic (e.g., Figure 1A and 1B or a unified pathway diagram) that distinguishes the LSD versus 5-HT pathways using color-coded arrows (e.g., blue for LSD, orange for serotonin).

Figures 1 and 2 were combined in single figure (Figure 2), clear color distinction between arrows indicating induced pathways was created. Figure legend was accordingly changed. Appropriate changes were made in the text of the manuscript.

  • The pyramidal neuron illustration in these figures is misleading and should be removed. The 5-HTâ‚‚A receptor is not exclusive to pyramidal cells; rather, it is broadly expressed across multiple neuronal and even non-neuronal cell types throughout the brain and peripheral organs, as acknowledged by the authors themselves.

Removed

  • I strongly recommend adding a systems neuroscience-style figure that illustrates serotonergic projections from the raphe nuclei and the distribution of 5-HT receptor subtypes across different brain regions. Such a figure would help bridge intracellular signaling pathways with large-scale neural circuit dynamics, particularly relevant to the discussion starting at line 167.

A new figure (Figure 1) was included designating the neuronal pathways and areas in the brain affected by ascending serotoninergic pathways. Areas of the highest concentrations of 5-HT2A receptors were indicated. Brief description of the 5-HT influence over distinct brain regions was provided in the legend.

  • The role of 5-HTâ‚‚A receptors in sensory processing under normal physiological conditions remains underexplored. A brief summary of what is known from animal models or human studies would contextualize their dysfunction in SP and psychedelic states.
  • The manuscript’s goal—bridging molecular signaling with cognitive and perceptual states—would benefit from a dedicated paragraph linking molecular and circuit-level mechanisms. Consider expanding the discussion to include electrophysiological findings from animal and human studies. For instance, going beyond the brief mention of alpha oscillations and the N170 potential (line 362) would enrich the discussion, especially regarding network-level correlates of hallucinations.

Two new subchapters were included discussing the role of the 5-HT2A receptor in physiological sensory processing and the role of 5-HT2A in shaping perception. Included subchapters highlighted in yellow. Furthermore, to clarify the text, subchapter regarding serotonin was moved ahead of the psychedelic substances.

  • In a review, a section dedicated to “outstanding questions and unresolved issues” is welcomed. Please add a short paragraph highlighting what the authors think are the most pressing unknowns.

A new subchapter was included: “Unresolved questions and working hypothesis” before summary, where the unknowns are briefly discussed and a new working hypothesis is included alongside a potential model to verify its validity. Appropriate change was also included in the abstract of the manuscript. Changes are highlighted in yellow.

  • The Summary section would benefit from a more integrative perspective. Specifically:
    • Emphasize the temporal mismatch between molecular signaling (e.g., β-arrestin-2 pathway activation) and the fast, transient nature of hallucinations during SP.
    • Discuss how the electrical state of the 5HT2A-expressing neurons (e.g., depolarization thresholds, excitatory/inhibitory balance) might modulate susceptibility to hallucinations.

The summary section was enhanced.

Minor comments:

All inclusions, corrections or clarifications as well as new citations are highlighted in yellow.

  • Lines 49–50: Please add a reference supporting the claim that altered serotonergic signaling primarily disturbs pyramidal neuron function.

Included in the text

  • Line 85: the authors state “… not identical to those after ingesting psychedelic substances, some important similarities do exist” but fail to describe which ones. Please specify the nature of these similarities (e.g., visual distortions, altered sense of self or time, etc.).

The sentence was accordingly clarified

  • Line 94: The sentence on hypnagogic and hypnopompic phases adds little without further elaboration. Removing this sentence will improve clarity, but if retained, it should be better integrated into the paragraph and define both terms for clarity.

Removed

  • Line 98: “In specific regions…” > please name these brain regions explicitly. Ideally, they would be visualized in the proposed new figure.

Brain regions were designated in the text

  • Line 109: Clarify the “three-component model” of SP. If referring to intruder, incubus, and unusual bodily experiences, please list them explicitly and define as needed.

The term and its components were explained

  • Line 115: Provide a brief definition of autoscopy at first mention.

Definition of autoscopy was included in the sentence

  • Lines 123 onward: Numerous brain areas are cited (e.g., SPL, TPJ, right parietal cortex). It would be valuable if these regions were annotated in the suggested systems-level figure.

A new figure (Figure 1) was included in the manuscript.

  • Line 168: Add a citation to substantiate the claim that these regions “play a key role in regulating muscle tone and processing sensory information.”

References were included in the text

  • Line 177: Define synesthesia briefly.

Definition of synesthesia was included in the text

  • Line 208: Add reference (perhaps reference #6) for the mechanism discussed.

Reference was designated, indeed it was reference 6

  • Line 224 & 226: Add appropriate citations for claims regarding the role of these substances in cognitive/mood alterations and their essential role in hallucinogenesis.

References were included in the text

  • Lines 254 onward: The introductory paragraph of this section is a dense, single sentence that is difficult to follow. Please rephrase for clarity and introduce the concept of biased agonism with a brief explanation at first use.

Paragraph was rewritten to clarify its meaning.

  • Lines 272 onward: Specify which figure is being referenced (presumably Figure 2) and ensure figures are cited appropriately throughout.

Clear indications to figure 2, part A and B according with references to differentially colored pathways were included in the text of the manuscript.

  • Table 1: Please adopt a consistent naming convention for the column “compound”, such as Chemical name (common name) or vice versa.

The consistent nomenclature was adopted with chemical name, followed by abbreviated symbol with a common name / abbreviation in the parentheses

  • Line 317: The phrase “in this review…” is misplaced, as it occurs near the end of the article. Consider removing or rephrasing.

Rephrased

  • Line 355: Define psychotomimetic.

Psychotomimetic effects were explained in the text

  • Line 362: Define N170 as the negative deflection in local field potential ~170 ms after a visual stimulus.

Explained in the text

Round 2

Reviewer 1 Report

Comments and Suggestions for Authors

No additional comments

Reviewer 2 Report

Comments and Suggestions for Authors

I appreciate the authors' effort to enhance the quality of the manuscript and to satisfactorily answer all of my comments.  Congratulation for this fine review.